# Homogeneity Measurements of Li-Ion Battery Cathodes Using Laser-Induced Breakdown Spectroscopy

**DOI:** 10.3390/s22228816

**Published:** 2022-11-15

**Authors:** Moritz Kappeler, Carl Basler, Albrecht Brandenburg, Daniel Carl, Jürgen Wöllenstein

**Affiliations:** 1Department of Production Control, Fraunhofer Institute of Physical Measurement Techniques IPM, Georges-Köhler-Allee 301, 79110 Freiburg, Germany; 2Department of Gas and Process Technology, Fraunhofer Institute of Physical Measurement Techniques IPM, Georges-Köhler-Allee 301, 79110 Freiburg, Germany; 3Laboratory for Gas Sensors, Department of Microsystems Engineering, University of Freiburg, Georges-Köhler-Allee 102, 79110 Freiburg, Germany

**Keywords:** laser-induced breakdown spectroscopy, atomic emission spectroscopy, Li-Ion battery, binder, conductive additive, inline process monitoring

## Abstract

We study the capability of nanosecond laser-induced breakdown spectroscopy (ns-LIBS) for depth-resolved concentration measurements of Li-Ion battery cathodes. With our system, which is optimized for quality control applications in the production line, we pursue the goal to unveil manufacturing faults and irregularities during the production process of cathodes as early as possible. Femtosecond laser-induced breakdown spectroscopy (fs-LIBS) is widely considered to be better suited for depth-resolved element analysis. Nevertheless, the small size and intensity of the plasma plume, non-thermal energy distribution in the plasma and high investment costs of fs-LIBS make ns-LIBS more attractive for inline application in the industrial surrounding. The system, presented here for the first time, is able to record quasi-depth-resolved relative concentration profiles for carbon, nickel, manganese, cobalt, lithium and aluminum which are the typical elements used in the binder/conductive additive, the active cathode material and the current collector. LIBS often causes high variations in signal intensity from pulse to pulse, so concentration determination is, in general, conducted on the average of many pulses. We show that the spot-to-spot variations we measure are governed by the microstructure of the cathode foil and are not an expression of the limited precision of the LIBS setup.

## 1. Introduction

Due to their high specific energy density and cycling stability, Li-Ion batteries are one of the most popular technologies for storing electrical energy. Although degradation mechanisms are unavoidable in the current state of research, bad operating conditions (low temperature, deep-discharging, high charging/discharging-currents) as well as unfavorable production circumstances can significantly accelerate aging. A variety of, in part, complex mechanisms for the aging of Li-Ion batteries have been described in the literature and manifest themselves electrochemically as capacity fade, impedance rise and overpotential [1]. Although electrochemical models for the half-cells [2] in combination with charge and discharge curves allow a further classification into different degradation modes, the specific microscopic mechanism can still be versatile [3]. A different approach focusing on the underlying physical and chemical principles of battery aging is the elemental analysis of lithium-ion battery constituents and their degradation products. Various X-ray-based analytical methods have been applied and are covered by several reviews [4,5,6]. As summarized in [7], a variety of other analytical techniques, including mass spectroscopic methods (ICP-MS, SIMS), Atomic Absorption Spectroscopy (AAS), Raman Spectroscopy and Optical Emission Spectroscopy (GD-OES, ICP-OES and LIBS) have been used to monitor the transition metal dissolution (TMD) into solvent and its accumulation on the anode side. The electrolyte itself has been extensively analyzed by ion chromatography and gas chromatography-mass spectroscopy (GC-MS) for salts and acids which could affect the dissolution of transition metals. ICP-OES and AAS have also been successfully applied to detect Li loss in discharged anode material (graphite) by forming Li^+^ vacancies [8] and to monitor the lithium distribution after extensive cycling [9].

LIBS outperforms other analytical techniques, such as X-ray fluorescence, by higher sensitivity for light elements. It is fast compared to Raman spectroscopy and GD-OES, and inline-capable in contrast to EDX and SIMS. LIBS furthermore allows the detection of a large number of different elements simultaneously, provided that microscopic damage can be tolerated. In 2012, Zorba et al. [10] studied the chemical composition of the solid electrolyte interface with respect to lighter elements using fs-LIBS with 7 nm depth resolution. Moreover, Imashuku et al. [11] showed the possibility of quantitative lithium mapping and detection of electrolyte decomposition products on cycled cathodes using time-resolved ns-LIBS. In 2019, depth-resolved fs-LIBS has also been demonstrated on a laser-structured 3D cathode architecture for the investigation of Li distribution [12].

In this study, we apply time-integrated ns-LIBS for quasi-depth-resolved elemental mapping of C and Al with respect to active material. Emphasis is put on the imaging of conductive additives (graphite and carbon black) for applications in production control.

## 2. Materials

The cathode of Lithium-ion batteries used in this study is composed of a Lithium Nickel Manganese Cobalt oxide (NMC) as an active material with a layered atomic structure (LiNi_x_Mn_y_Co_z_O_2_, x + y + z = 1) which allows for the intercalation of Li^+^ ions, a current collector made of aluminum, Polyvinylidene fluoride (PVDF) as a polymer binder to ensure mechanical stability and conducting additive (graphite and carbon black) to ensure electrical conductivity to the current collector. The cathode material examined in this paper was produced by Fraunhofer ISIT and consists of an active material with composition x = 1/3, y = 1/3, z = 1/3, abbreviated NMC 111, and different amounts of additives (graphite/carbon black between 0 and 15% and binder between 6 and 11%). To produce the cathodes, PVDF was first mixed with acetone and dissolved. The solution was then mixed with the conductive additive and finally homogenized by ultrasonic dispersion. The concentrations of the different constituents were controlled gravimetrically during the production of the slurry. The slurry was applied to the aluminum foil with a doctor blade at a constant coating speed. Evaporation of the slurry on the coated current collector took place at room temperature.

## 3. Methods

The Zentraleinrichtung Elektronenmikroskopie (ZELMI) of the technical university of Berlin used electron microprobe analysis (EMPA) to determine the particle geometry of active material, its distribution and chemical composition along a cleaved cross-section of an NMC 111 cathode. This measurement should deliver information on the spatial arrangement of the NMC particles and their vacancies and complement the information on concentration known from gravimetrical measurement on the macroscopic dimension.

Optical emission spectroscopy, using a self-built, inline-capable ns-LIBS system, was carried out with a variety of cathodes with different compositions. Figure 1 shows a simplified sketch of the setup. A diode-pumped Nd:YAG laser (Quantum Light Instruments Q2-100) with 1064 nm wavelength, a pulse duration of 6 ns, 1–100 Hz pulse repetition rate, a pulse energy between 0.25 and 50 mJ, and pulse-to-pulse energy stability < 0.5% (RMS) serves as an excitation source and is focused to a spot with a waist size of approximately 50 µm. The output power of the laser was measured using an optical energy meter with a pyroelectric sensor (Newport 818E-03-12L) While fs-LIBS is generally preferred for depth-resolved concentration measurements compared with ns-LIBS, the larger plasma plume and higher signal intensity of our nanosecond laser system turned out to be superior regarding inline applications. The small plasma plume of the fs-LIBS needed very accurate dynamic positioning with respect to the moving cathode foil and could not guarantee the necessary robustness needed for inline applications.

In favor of a larger optical throughput, three Czerny-Turner-type Avantes StarLine spectrometers with fixed grating were preferred over Echelle types, covering the whole region of interest (193 nm–671 nm). The devices are equipped with second-order filters to suppress the secondary spectrum. In all three spectrometers, a slit size of 25 µm was used. For the UV range (193 nm to 262 nm) the spectrometer is equipped with 2400 lines/mm grating, giving a wavelength resolution < 170 pm. For the UV-VIS range (268 nm–536 nm) and the VIS region (500–736 nm), the spectrometer comes with 1200 lines/mm gratings. A resolution of < 400 pm was achieved in both cases. Exposure was synchronized with the laser and started right before the Q-switch was triggered. An exposure time of 2 ms close to the lower boundary of the device was chosen. Shorter exposures would have been desirable to reduce noise since delays of exposure showed that only the first 5 µs after the laser pulse contributed notably to the intensity of the optical spectrum.

Optical paths for excitation and detection are mechanically fixed and can be moved along the xy-plane with two motorized stages (OWIS LIMES 150, OWIS HPL 84N), the latter (and faster) with the optional feature to synchronize with a conveyor belt for inline applications. The fixed height of the sample was adjusted with an accuracy of approx. 200 µm over the whole measurement area of 100 mm × 80 mm. A triangulation sensor (Micro-Epsilon ILD 1320), close to the measurement spot, supervised the distance between the measuring head and sample to simplify the adjustment and exclude measurement errors due to misalignment.

## 4. Results

### 4.1. Cathode Characterization with EMPA

Backscattered electron microscopy has been used to obtain high-resolution images (Figure 2) of the shape and location of NMC particles.

Using OpenCV’s threshold and contour methods, the locations and sizes of the particles were extracted. We determined a mean particle radius of 3.2 ± 1.7 µm (Figure 3) and a degree of coverage of 38%. This radius may differ from the real particle’s radius since the slice does not cut through every particle in the layout of the maximum cross-section.

The statistical information on NMC particles is essential to interpret spot-to-spot variations in the LIBS measurement. Both limited LIBS-based precision and the granular structure of the sample contribute to spot-to-spot variations in the measured LIBS signal. A detailed quantification of both contributions is necessary to distinguish between expected fluctuations and production-based irregularities. A detailed analysis will be presented in Section 4.4.

Wavelength-dispersive X-ray spectroscopy (WDX) data supplement the geometrical information with chemical contrast. The WDX signals for Ni, Mn and Co are shown in Figure 4. Whereas the chemical composition appears homogenous throughout a single particle, slight variations in composition were observed between different particles.

In the case of the binder, which can be identified by the WDX signal of fluorine, accumulations were observed around the surface of the NMC particles, forming rims in the raster image (Figure 5).

Since both the binder and graphite/carbon black, contain carbon, the distinction between binder and graphite/carbon black is not straightforward. The structure seen in Figure 5 can also be seen in the WDX signal for carbon (Figure 6) but superimposed by stronger contributions from the vacancies between different NMC particles, which we assume to be attributed to graphite/carbon black.

The LIBS analysis typically covers a round area of 50 µm waist size and the carbon fluctuations viewed in Figure 6 would appear smoothed in a single measurement. Averaging is furthermore beneficial due to the limited precision of LIBS in general, but the destruction of the sample at the measurement spot makes repetition impossible at the same position.

### 4.2. LIBS Measurement

The Czerny–Turner-type spectrometers with mechanically fixed grating cover the whole spectrum of interest for C, Ni, Co, Mn, and Li (193 nm to 671 nm). Corresponding spectra for NMC cathode material are shown in Figure 7a–c. A small mismatch was observed by comparison of the observed lines with the NIST database [13]. They are −40 pm (Carbon line) to −100 pm (Manganese line) for the UV device (186 nm–253 nm), −500 pm to –200 pm for the UV-VIS device (270 nm–530 nm) and +400 pm to +700 pm for VIS device (500 nm–730 nm).

Atomic spectral lines of C have been observed from the graphite or binder component. Many strong transitions in the vacuum UV region are present according to spectroscopic databases, but not measurable with a fiber spectrometer and in a LIBS experiment under atmospheric conditions. Nevertheless, single lines in the measurable spectral region at 193.0 nm and 247.9 nm were associated with Carbon.

Molecular bands had been observed in experiments with another laser system in our lab using fs pulses and were mainly attributed to radicals like diatomic carbon, cyano radicals and methylidyne radicals known from the Swan system [14], initially observed in 1856 [15]. We noticed that molecular bands and spectral lines were in focus for different adjustments of the detection system, indicating a spatial separation of the different species, as already observed by other groups [16]. The simultaneous presence of molecular bands and atomic lines made the evaluation of the time-integrated LIBS signal challenging with this laser system.

The major spectral lines of neutral Li are found in the red region of the visible spectrum, namely 670.8 nm (2p → 2s transition) and 610.4 nm (3d → 2p transition) [13]. The former transition occurs between two states of the same shell which are strongly split due to different Coulomb interactions of the different orbitals to the core electrons and nucleus [17]. This spectral line suffers from self-absorption [18] in case of high concentrations which can lead to wrong quantification results, if not accounted for [19].

The optical emission spectrum of Ni in UV is dominated by transitions of the outmost electrons, namely the eight d electrons and the two outer s electrons. The strongest lines either involve the ground level configuration 3d^8^(3F)4s^2^ or the 3d^9^(2D)4s configuration. A high density of states is found in the range 3.3 eV–4.2 eV (spectral lines between 280 nm and 400 nm), around 5.3 eV–5.6 eV (spectral lines in the range 228 nm–242 nm) and for energies larger than 6 eV [13].

Similar to Ni, the Co emission in the visible spectrum is governed by transitions of the seven d electrons and the outmost two s electrons. Several different terms for the configurations 3p^6^3d^7^4s^2^, 3p^6^3d^8^4s and 3p^6^3d^8^4s4p are responsible for a great number of transitions mainly in the UV spectrum shorter than 300 nm.

For Mn, excited states are lifted at least 2.1 eV from the ground level. A high density of states is found in the range 2.1–2.3 eV, 2.9–3.4 eV and 3.7–3.9 eV. For energies > 4.2 eV many states are registered. Most of the prominent lines involve either the ground level (distinct lines at 279 nm, 322 nm, 403 nm, 539 nm) or a lower level in the range 2.1–2.3 eV (lines in the ranges 304–307 nm, 320–327 nm, 353 nm–373 nm, 379 nm–385 nm and 404 nm–408 nm) [13].

For Al, NIST data [13] show a large separation of 3.1 eV between the ground state and the excited states. Above 4.8 eV, the density of states is highly increased. Several terms at 3.1 eV, 3.6 eV, 4.1 eV and 4.6 eV are responsible for strong lines around 395 nm, 309 nm, 257 nm and 265–266 nm in the UV spectrum. Although detected in the spectrum, ionic Al II lines turned out to be much more intense compared to Al I lines and were subsequently used. The most prominent line of Al II detected in our measurement was between 198 nm and 199 nm.

It is worth mentioning that spectral features were observed around 185 nm–193 nm, which we ascribe to Schumann-Runge bands of O_2_ [20].

### 4.3. Concentration Measurement

For quantitative analysis, the LIBS system was calibrated on NMC 111 samples with different mass concentrations of graphite/carbon black (0–15%) and binder (6–11%). We observed an increase in the intensity of carbon lines with respect to metallic lines for increasing pulse energy in the range below 2 mJ, which potentially indicates non-stoichiometric ablation [21]. Calibration measurements were performed at 6 mJ, in the stable region above 3.5 mJ.

The large fluctuation in the intensity of spectral lines between successive laser pulses makes quantitative analysis directly from the line intensity challenging. This problem is often met with normalization methods, e.g., internal standard [22], standard normal variate of the whole spectrum, area under the spectrum, or background. Several publications treat the problem of selecting appropriate lines such as avoiding or dealing with resonant [23] and self-absorbing lines of major components. The selection of lines with similar upper energy levels can minimize the influence of changing plasma parameters [24]. For C, both observed lines in the UV spectrum, 193 nm and 248 nm share the same upper energy level (2s^2^3p3s (1P)) of 7.68 eV. This is comparable with the first ionization energy of Co (7.88 eV), Ni (7.61 eV) and Mn (7.43 eV), and therefore, larger than the typical upper energy levels of intensive lines in the spectrum from NMC targets [13]. Taking the intensity ratio of two spectral lines with different upper energy levels makes the measurement plasma temperature dependent. Nevertheless, reproducible calibration curves were achieved with normalization to the standard normal variate of the spectrum which we did for all evaluations shown in the following.

A strong correlation between the 193 nm as well as the 248 nm carbon line with the graphite/carbon black concentration was observed (Figure 8a, Pearson correlation coefficient R^2^ = 0.997), whereas the concentration of binder had no significant influence on the line intensity of C (Figure 9a). This could have several reasons: 1. Non-stoichiometric ablation of the sample, i.e., only little amount of the binder enters the plasma, whereas graphite/carbon black is easily ablated. 2. Long-chained polymer binder rather forms molecules instead of atomic species [25]. 3. Different plasma conditions, e.g., lower plasma temperature in the polymer containing plasma, prevent the upper energy level corresponding to the 193 nm and 248 nm line to be occupied.

For each of the six samples, 25 measurements were taken at each of the 25 measurement spots. The average was taken over all measurements at the same spot. From those 25 spectra, the C signal was evaluated from the spectral region between 192.8 nm and 193.3 nm (247.7 and 248.0 nm) by dividing the baseline-corrected area through the standard normal variate of the whole spectrum. Figure 8b shows calibration curves together with the point-to-point deviation of the LIBS signal illustrated by error bars. Besides the growing standard deviation with increasing graphite/carbon black content, a significant Y offset has been observed which can hardly be explained by interference with other spectral lines. On the other hand, as already mentioned above and shown in Figure 9b, a variation of binder shows no significant influence on either the carbon signal. This offset could be due to residues from the solvent remaining in the sample. Further investigations are needed to identify the cause.

Detailed statistics on spot-to-spot signal variations were gained from a 31.5 mm × 20 mm measurement field with 2520 measurement spots and 60 laser pulses at each position (Figure 10).

The mean spot-to-spot standard derivation inside each layer is 27.3%. We will show in the next section that we expect a similar value due to the inhomogeneity of our cathode sample.

A decrease in Carbon signal has been observed systematically for subsequent laser pulses at the same spot (Figure 11). Since this decrease is again independent of the binder concentration, we conclude that it is related to the ablation of graphite/carbon black. We interpret this observation as a result of a lower ablation threshold of graphite/carbon black compared to other constituents and conclude that uniform depth profiling is challenging with this composition of the material.

Nevertheless, reproducible depth profiling is possible with some limitations: As illustrated in Figure 12a, the aluminum current collector is uncovered with subsequent laser pulses leading to a strong increase in the single-ionized aluminum (Al II) line at 198.8 nm. The ablation rate significantly changes with laser power. The Al II signal is viewed in Figure 12b for subsequent laser pulses with 6 mJ energy per pulse at four different positions.

Although the evolution of the Al II line showed good agreement for many different measurement points, a comparison with microscopic images pointed out, that the ablation rate significantly depends on the particle distribution at the specific position. The presence of large NMC particles slows down the uncovering process, as shown in Figure 12b (position 4), while the ablation rate of binder and graphite turns out to be higher. This leads to protruding NMC particles around the already vaporized binder and graphite/carbon black areas at the measurement spot. Additionally, the Gaussian profile of our laser beam together with the thermal conductivity of the sample leads to a V-shaped crater, which makes a difference between the profiles shown in Figure 12b and a step function. We conclude, that under ideal conditions, i.e., controlled distance to sample and constant laser power, our system is able to detect relative depth variations in the order of 20% for depths in the range of 25–100 µm.

### 4.4. Cathode Characterization with EMPA

Given that a single LIBS pulse takes the average concentration inside the focal area of the excitation beam, the measurement can be understood as a smoothing operation of the local concentration within the beam’s waist size w. We consider the smoothing kernel:(1)$$h\left(x,y\right)=\frac{2}{\pi w^{2}}\exp\left[-\frac{2\left(x^{2}+y^{2}\right)}{w^{2}}\right],$$

Its spatial Fourier transform corresponds to the transfer function and is given by:(2)$$H\left(k_{x},\;k_{y}\right)=\int_{-\infty }^{\infty }\left[\int_{-\infty }^{\infty }h\left(x,y\right)\;\exp\left(ik_{y}y\right)dy\right]\exp\left(ik_{x}x\right)dx=\exp\left[-\frac{\left(k_{x}{}^{2}+k_{y}{}^{2}\right)w^{2}}{2}\right],$$

The transfer function quantifies how spatial frequencies of concentrations are “smoothed out” by averaging over the measurement spot. When fluctuations are taking place on small length scales compared to the laser spot size, the measured variations are smoothed out efficiently by the extent of the laser spot. It is, therefore, instructive to quantify the spatial frequencies of local concentrations by Fourier transform and compare it to the transfer function *H*.

For the spatial carbon distribution of Figure 6, we obtain the discrete Fourier transform as illustrated in Figure 13 with a logarithmic scale.

The center of Figure 13 represents the average concentration of the whole measurement area whereas the points around correspond to amplitudes of concentration variations with different spatial frequencies. The pixels in Figure 13 are not square due to different pixel numbers in the x and y directions in Figure 6. The smoothing operation suppresses higher frequencies by multiplying the data pointwise with *H* (shown in Figure 14). The radius of the transfer function’s spot is inversely proportional to the laser spot’s radius. After smoothing, the squared sum of remaining amplitudes for frequencies other than (0 µm^−1^,0 µm^−1^) represents the expected amplitude of fluctuations.

For the chosen dataset, the relative amplitude of fluctuations is compared for different laser spot sizes in Figure 15.

We, therefore, expect a strong dependency between laser spot size and signal variance for this kind of inhomogeneous sample. For a laser waist size of 50 µm, we obtain an expected relative variance of 27–32% which is in good agreement with the data shown in Figure 6 Thus, the laser spot radius appears as an important parameter when measuring inhomogeneous samples and needs to be compared to the typical length-scale of the structure.

## 5. Discussion and Conclusions

We examined the potential of nanosecond laser-induced breakdown spectroscopy (ns-LIBS) for depth-resolved concentration measurements in lithium nickel manganese cobalt oxide (NMC) cathodes for Lithium-Ion batteries. We preferred ns-LIBS over fs-LIBS because of the larger plasma plume, higher signal intensity and lower investment costs which we consider crucial for industrial inline applications. Although ns-LIBS is generally considered less suitable for depth-resolved concentration analysis, our measurements show repeatable intensity evolution of normalized spectral lines from the current collector for successive laser pulses under a well-controlled laser and focusing conditions. Despite the fact that LIBS always suffers from pulse-to-pulse fluctuations, in the case of NMC cathodes, we could show that the pulse-to-pulse fluctuations can be explained by the homogeneity of the sample itself. Thus, we conclude that LIBS is capable of spatially resolved concentration measurements even on very inhomogeneous samples. Our study on depth-resolved carbon detection, on the other hand, indicated non-uniform ablation of the different components in the material composition which makes depth-profiling challenging: Large particles at the specific measurement points slow down the ablation process since the ablation rate of NMC has been observed to be lower compared to graphite/carbon black and binder. Additionally, the ablation of carbon was observed to be disproportionately high in the beginning and dropped for successive laser pulses at the same measurement spot. We noticed relatively large fluctuations of 30% in C signal between single LIBS spectra at different measurement spots which we could ascribe to the microstructure of the cathode by comparison with electron microprobe analysis. If concentration fluctuations are on the length scale of the laser spot size, they strongly impact the LIBS measurement. We showed that, despite this effect, an accurate concentration measurement of C is possible by applying an average procedure with a sufficient number of N of single measurements at different positions. This reduces the relative standard variations with respect to a single measurement according to N^−1/2^. Besides averaging, we suggest for inhomogeneous samples to increasing the laser spot size in order to smooth out local concentration variations.

## Figures and Tables

**Figure 1 sensors-22-08816-f001:**
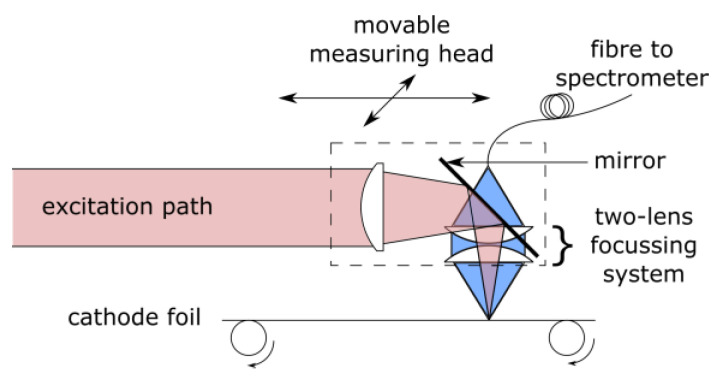
Inline-capable ns-LIBS measurement setup. Red: optical excitation path for 1064 nm Nd:YAG laser with active Q-switch, blue: optical detection path (behind excitation path) with two-lens collimating and refocusing system which couples the optical emission into a fiber. Both optical paths share the same focus point.

**Figure 2 sensors-22-08816-f002:**
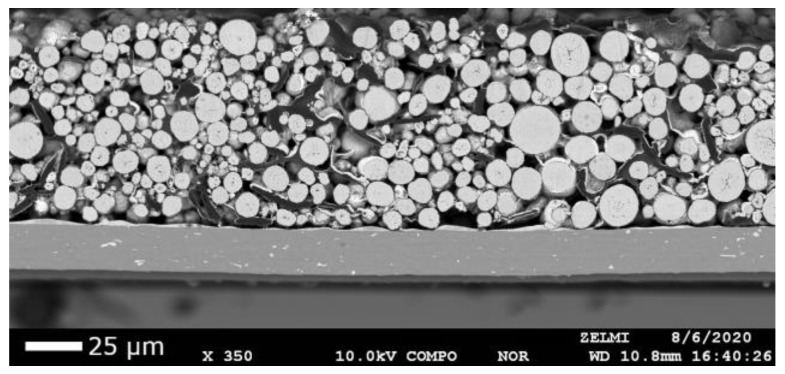
High-resolution image of a positive NMC 111 lithium-ion battery cathode obtained by the backscattered electron signal of a microprobe analysis, acceleration voltage: 10 kV.

**Figure 3 sensors-22-08816-f003:**
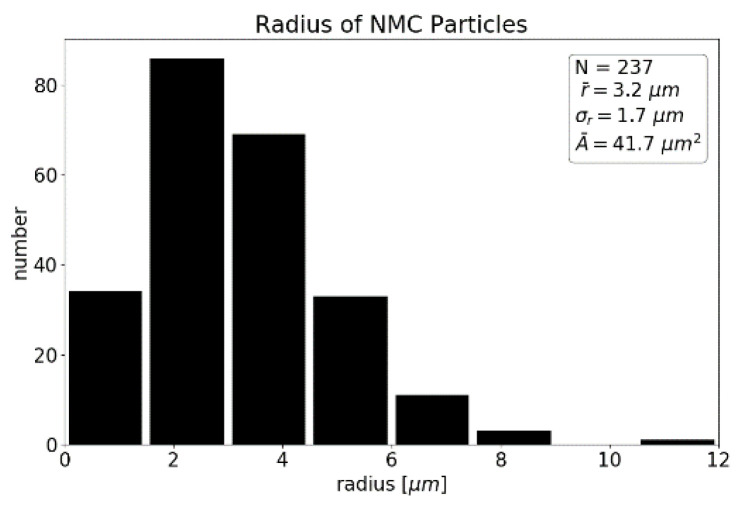
Histogram of particle radii of an NMC 111 lithium-ion battery cathode from the Blob analysis.

**Figure 4 sensors-22-08816-f004:**
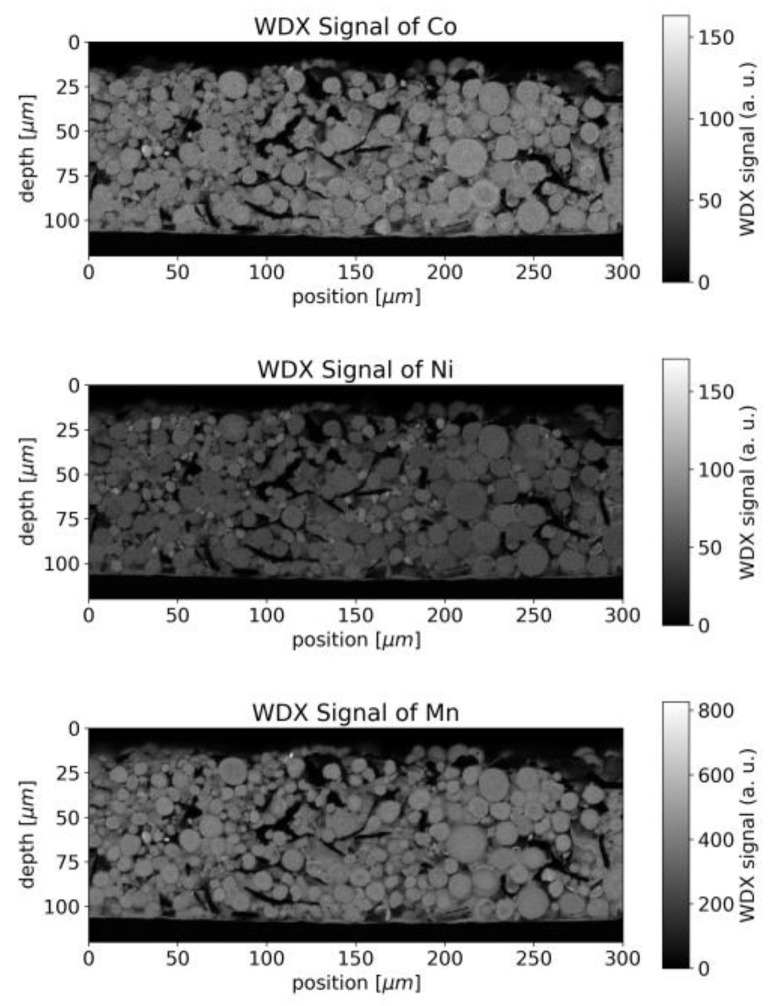
WDX signal of active materials from EMPA measurement for NMC 111 of Co, Ni and Mn.

**Figure 5 sensors-22-08816-f005:**
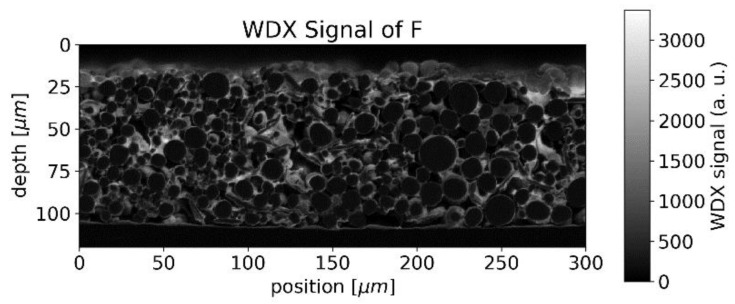
WDX signal of fluorine from EMPA measurement (indicating binder).

**Figure 6 sensors-22-08816-f006:**
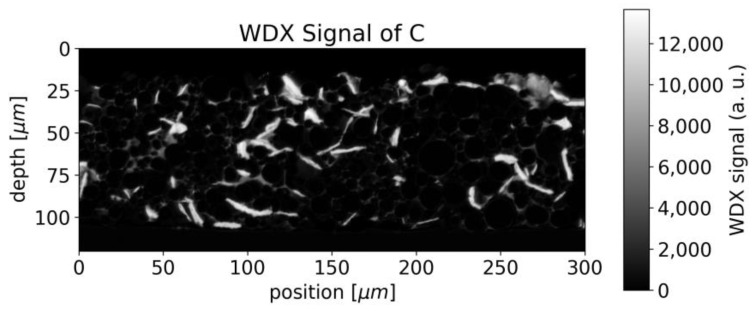
WDX signal for carbon from EMPA measurement (indicating graphite and binder).

**Figure 7 sensors-22-08816-f007:**
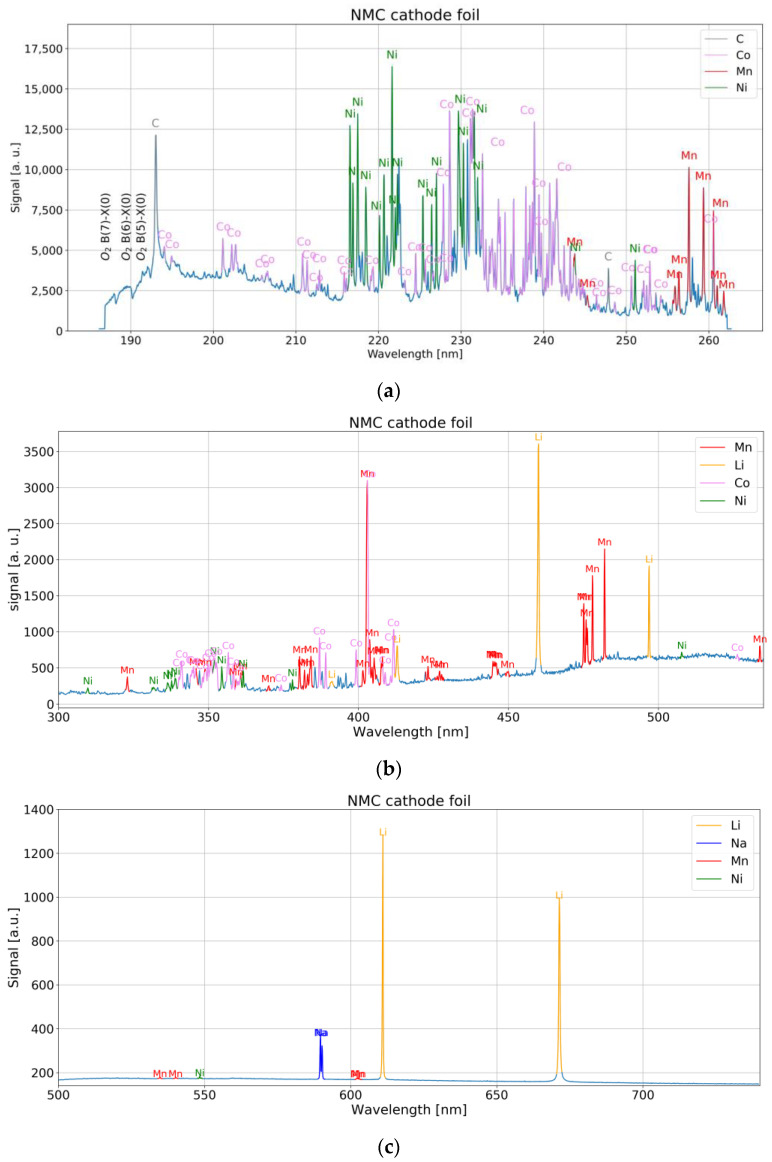
(**a**) UV spectrum of NMC cathode foil with 6 mJ pulse energy. (**b**) Near-UV-VIS spectrum of NMC cathode foil with 1 mJ pulse energy. (**c**) VIS spectrum of NMC cathode foil with 1 mJ pulse energy.

**Figure 8 sensors-22-08816-f008:**
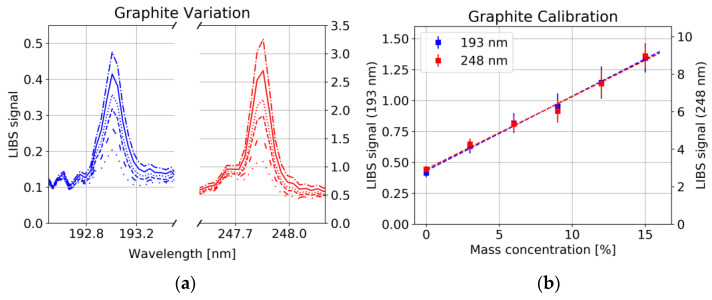
(**a**) Carbon lines (std normalized) for different concentrations of graphite/carbon black (NMC111, 6 mJ) and 8% binder, each spectrum is the average of 625 single measurements. (**b**) Calibration curves for graphite/carbon black content error bars from 25 samples (25 spectra averaged for each sample).

**Figure 9 sensors-22-08816-f009:**
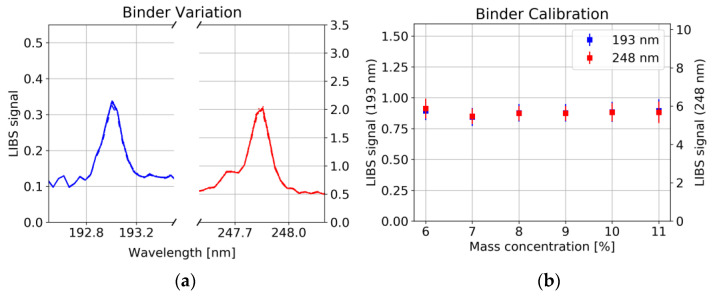
(**a**) Carbon lines (std normalized) for different concentrations of binder (NMC111 6 mJ) and 6% graphite/carbon black. (**b**) LIBS signal for six samples with different Binder concentrations, error bars from 25 samples (25 spectra averaged for each sample).

**Figure 10 sensors-22-08816-f010:**
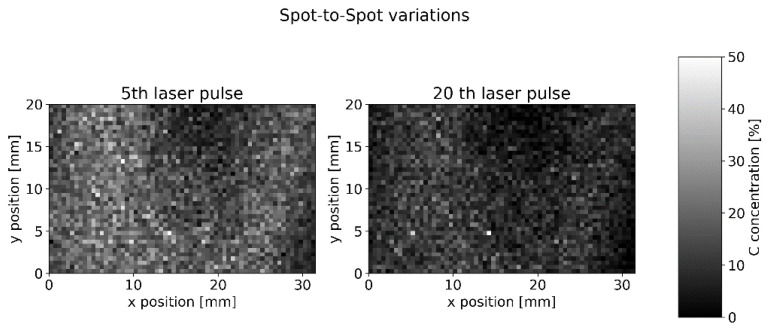
Single ns-LIBS measurement of C concentration (193 nm line) on NMC 111.

**Figure 11 sensors-22-08816-f011:**
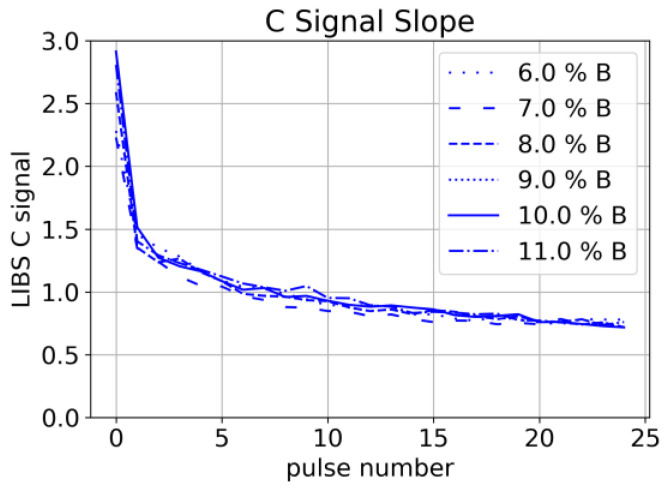
Slope of C signal (193 nm line) for subsequent pulses at the same measurement spot for samples with different Binder concentrations (each sample: mean of 25 spectra).

**Figure 12 sensors-22-08816-f012:**
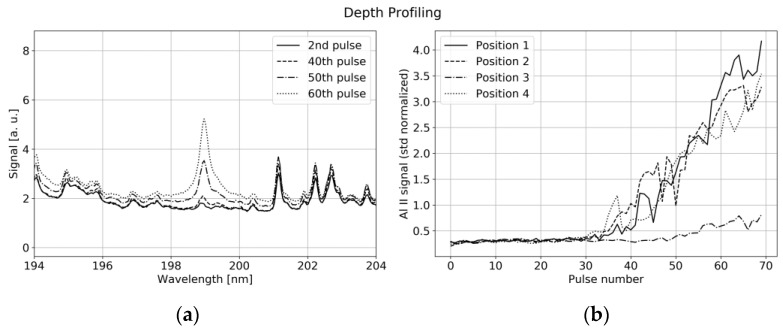
(**a**) VUV spectrum around a prominent Al II ionic line for subsequent laser pulses at the same position with 6 mJ pulse energy. (**b**) Signal of Al II ionic line for subsequent laser pulses at four different points along a line.

**Figure 13 sensors-22-08816-f013:**
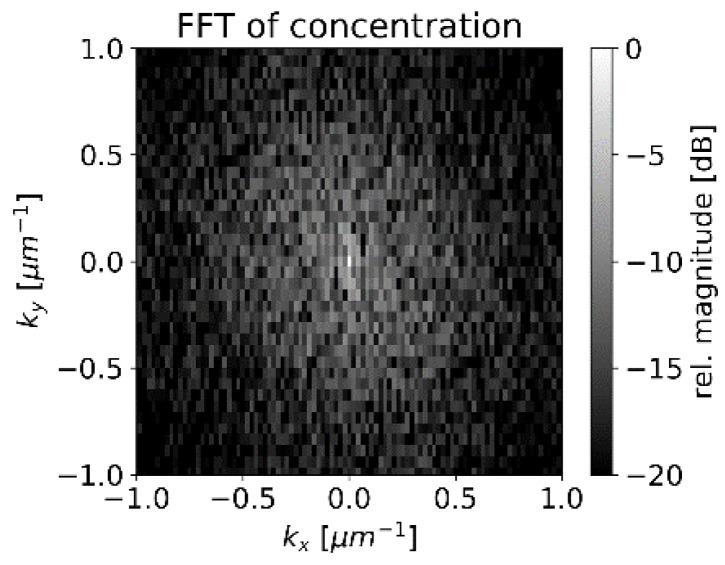
Small section of the discrete Fourier transformation of Carbon data (Figure 6) with the dominant frequencies.

**Figure 14 sensors-22-08816-f014:**
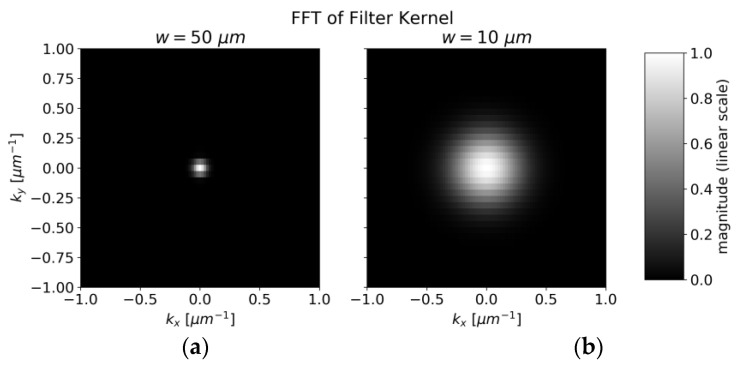
(**a**) Transfer function H(k_x_, k_y_) for 50 µm waist size of the excitation beam. (**b**) Transfer function for 10 µm waist size.

**Figure 15 sensors-22-08816-f015:**
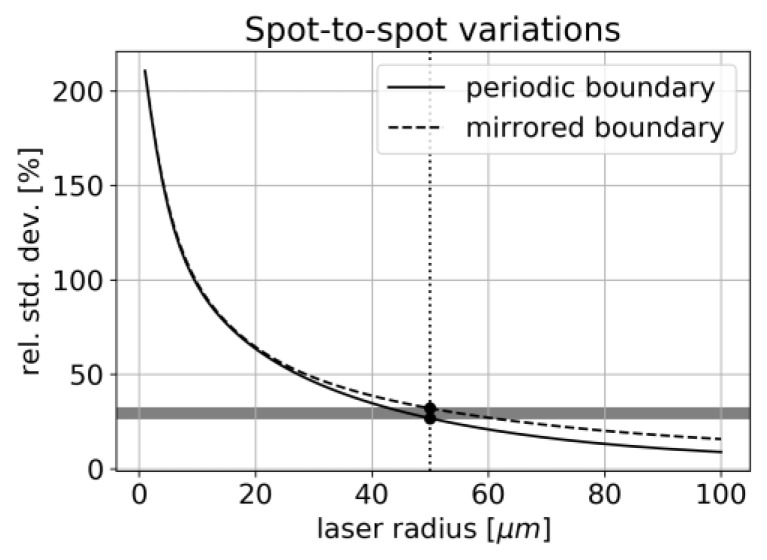
Expected spot-to-spot signal variation for cathode sample.

## Data Availability

Not applicable.

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
