# Peer review of "Homogeneity Measurements of Li-Ion Battery Cathodes Using Laser-Induced Breakdown Spectroscopy"

_sensors, 2022, doi:10.3390/s22228816_

Round 1
Reviewer 1 Report
In this study real-world example of LIBS depth profiling is presented on Li-Ion battery cathods which is quite difficult task because this sample is not homogenous. however, authors present clear data and also justification why fs-LIBS was not used in this study. Paper is clearly written with sound data and conclusions. Authors should correct sentence in the Abstract section starting at row 17 - "Nevertheless ..." , high investments costs can not make ns-LIBS more atractive...
Author Response
Dear Reviewer,
thank you very much for your revision of your manuscipt. I like to comment your suggestion to improve the introduction: "Authors should correct sentence in the Abstract section starting at row 17 - "Nevertheless ..." , high investments costs can not make ns-LIBS more atractive..."
This sentence is in the context of the previous sentence, that ns-LIBS is much cheaper than fs-LIBS.
To make it more clear, we can put it in the following way:
"Nevertheless, the small size and intensity of the plasma plume, non-thermal energy distribution in the plasma and high investment costs of fs-LIBS make ns-LIBS more attractive for inline application in the industrial surrounding."
Does this adress your comment correctly?
New version attached.

Reviewer 2 Report
The authors presented nanosecond laser induced breakdown spectroscopy study for depth-resolved concentration measurements of lithium-ion battery cathodes. Although the topic has been previously extensively studied by different experimental techniques, this study brings novelty to the research in Li-Ion batteries. The cathodes were manufactured internally of equal composition of NMC oxides with Li. An inline-capable ns-LIBS system, with diode pumped Nd-YAG laser was used to study a variety of cathodes with different compositions. The system is characterized with large plasma plume and high signal intensity what contributed much to inline application. Three spectrometers with fixed grating were used in order to cover the whole region of wavelengths from193 nm to 736 nm. The structure of the cathode was characterized by high-resolution images produced in electron microprobe analysis (EMPA) complemented by the wavelength-dispersive X-ray spectroscopy data. This analysis makes an indispensable for the interpretation of spot-to-spot variations in the LIBS measurements. Optical spectra of all major atomic lines of Li, Ni, Mn, Co and C are presented in three spectral regions and compared with NIST data. I wonder why the authors didn’t use the latest online version of NIST database in ref. [13] that can be properly cited as recommended on NIST web site. As the authors rightly observed, the simultaneous presence of molecular bands and atomic lines in spectra made the evaluation of the time-integrated LIBS signal challenging but they have performed suitable depth-resolved concentration analysis and showed a repeatable intensity evolution of normalized spectral lines from the current collector for successive laser pulses. The authors showed a strong dependency between laser spot size and signal variance for inhomogeneous samples and concluded that for a laser waist size of 50 µm they expect relative variance of 30 %
Author Response
Dear Reviwer,
thank you very much for reviewing our manuscript. Indeed we used the newer version of the NIST database. We changed the reference 13 and deleted reference 25 as this was duplicted (see attachment).
